# Inflammatory Mechanism of *Brucella* Infection in Placental Trophoblast Cells

**DOI:** 10.3390/ijms232113417

**Published:** 2022-11-02

**Authors:** Yu Xiao, Mengjuan Li, Xiaoyi Guo, Hui Zeng, Xuehong Shuai, Jianhua Guo, Qingzhou Huang, Yuefeng Chu, Bo Zhou, Jake Wen, Jun Liu, Hanwei Jiao

**Affiliations:** 1The College of Veterinary Medicine, Southwest University, Chongqing 400715, China; 2State Key Laboratory of Veterinary Etiological Biology, College of Veterinary Medicine, Lanzhou University, Lanzhou Veterinary Research Institute, Chinese Academy of Agricultural Sciences, Lanzhou 730000, China; 3Changchun Veterinary Research Institute, Chinese Academy of Agricultural Sciences, Yujinxiang Street 573, Changchun 130102, China; 4Department of Microbiology and Immunology, University of Texas Medical Branch at Galveston, Galveston, TX 77555, USA; 5The Immunology Research Center, Medical Research Institute, Southwest University, Chongqing 400715, China

**Keywords:** *Brucella*, trophoblast, pathogenesis, inflammatory response, infection mechanism

## Abstract

Brucellosis is a severe zoonotic infectious disease caused by the infection of the *Brucella*, which is widespread and causes considerable economic losses in underdeveloped areas. *Brucella* is a facultative intracellular bacteria whose main target cells for infection are macrophages, placental trophoblast cells and dendritic cells. The main clinical signs of *Brucella* infection in livestock are reproductive disorders and abortion. At present, the pathogenesis of placentitis or abortion caused by *Brucella* in livestock is not fully understood, and further research on the effect of *Brucella* on placental development is still necessary. This review will mainly introduce the research progress of *Brucella* infection of placental trophoblast cells as well as the inflammatory response caused by it, explaining the molecular regulation mechanism of *Brucella* leading to reproductive system disorders and abortion, and also to provide the scientific basis for revealing the pathogenesis and infection mechanism of *Brucella*.

## 1. Introduction

Brucellosis is a widespread chronic infectious disease caused by *Brucella* that infects humans and mammals. It has had a major impact on human life, health and social production since it was first recorded [1,2]. People are often infected with *Brucella* after eating animal products such as unpasteurized milk and meat or coming into direct contact with infected animals. Many large livestock are also afflicted by Brucellosis, leading to reduced productivity and hampered development of the industry [3,4,5]. According to incomplete statistics, there are at least 500,000 cases of *Brucella* infection worldwide every year [6]. *Brucella* is a facultative intracellular bacterium which is Gram-negative and causes clinical signs such as headache, muscle pain, arthritis, osteomyelitis, placenta, miscarriage, and reproductive disturbance in the host. So far, as many as twelve *Brucella* species have been isolated and identified here, including six classical species (*B. melitensis, B. abortus, B. suis, B. ovis, B. canis* and *B. neotomae*) and six novel cognitive species (*B. ceti, B. papionis, B. inopinata, B. pinnipedialis, B. microti* and *B. vulpis*). Among them, *B. melitensis* is the main bacterial type that infects humans. *B. abortus*, *B. suis* and *B. canis* also have the same pathogenic potential to humans [7]. *Brucella* can infect a variety of mammals and new species such as *B. ceti* and *B. pinnipedialis* have been found, which reveals that *Brucella* is widespread and has the biological characteristics of multiple types of host populations [8,9,10].

As a temporary organ of mammals, the placenta plays a role in ensuring the normal development of the fetus. It is of great significance in biological reproduction along with the development and delivery of the fetus. The placenta is responsible for supplying nutrients, decomposing or metabolizing waste, secreting hormones to regulate fetal development, and protecting the fetus [11,12,13]. When *Brucella* infects the placenta, higher levels of inflammation and extracellular replication occur. In phagocytes, *Brucella* needs to evade its powerful killing function, and their struggle with each other eventually leads to chronic persistent intracellular infection [14]. Trophoblast cells of mother and fetus together form the complex structure of the placenta. On the one hand, trophoblast cells ensure the smooth implantation of the embryo into the uterus and facilitate the docking of the maternal vasculature with the fetus; on the other hand, they are mainly responsible for the blood-mediated exchange of nutrients and gases between the fetus and the mother. Some trophoblast cell subtypes (spongiotrophoblast cells in mice, canal-associated trophoblast giant cells, etc.) also provide hormones and cytokines to the maternal blood, guide the permanent remodeling of blood vessels, and optimize the vascular environment, thereby facilitating substance exchange between the fetus and the mother [15,16,17].

The mechanism of placental tissue inflammation caused by *Brucella* is currently poorly understood, but different experiments have shown that *Brucella* takes trophoblast cells as target cells and effectively replicates on its ER to induce ER stress [18,19,20,21]. In eukaryotic cells, the ER is primarily responsible for the biosynthesis, folding, and modification of secreted and transmembrane proteins, as well as functions for maintaining calcium homeostasis. When the physiological demand for protein folding increases, the ER cannot function properly in time due to its limited capacity, resulting in a misfolded or UPR in this organelle, which is known as ER stress [22]. ER stress is a common feature caused by infection of many pathogens, and bacteria and viruses activate the UPR by implementing disruption of ER function. During this process, the occurrence of ER stress and UPR often involves ATF6, PERK and IRE1 transmembrane receptors [23]. If the protein-folding processing capacity of the ER is insufficient to satisfy the demand leading to the accumulation of unfolded proteins, the three sensors will activate the UPR signaling pathway in response to these stimuli in order to restore the normal functioning of the ER [24,25]. They assist the ER by promoting the production of chaperones, helping proteins fold properly, and degrading misfolded proteins [26]. If the ER is stimulated more and more strongly or the condition does not improve for a long time, the host cell will start the apoptosis program [27]. It can be seen that the infection of trophoblast cells by *Brucella* eventually causes placental inflammation and ER stress are inseparable, and the interaction between ER and *Brucella* may be an important research direction [20].

It is necessary to study the physiological and pathological mechanism of *Brucella* infection, and the inflammatory mechanism of *Brucella* infection of trophoblast cells has not been comprehensively and systematically elucidated. This paper mainly analyzes and reviews this aspect and lays a foundation for further revealing the pathogenic mechanism of *Brucella*.

## 2. Biological Process of *Brucella* in Trophoblast Cells

### 2.1. Intracellular Transportation

The invasion of host cells by *Brucella* is a relatively complex physiological process (Figure 1). Similar to invading macrophages, *Brucella* enters into the cell with more concealed virulence factors [28] to form BCV and acquire early markers (EEA1 or Rab5), followed by getting the late endosomal lysosomal proteins (LAMP-1 and Rab7) [29,30]. BCV briefly interacts with early endosomes, late endosomes, and lysosomes, during which multiple virulence factors prevent BCV and lysosome fusion through different pathways, and BCV relies on the T4SS virB secretion system to reach the ER and form specialized *Brucella* proliferation chambers [31] containing replication-competent BCVs that subsequently lose LAMP-1 and acquire the chaperone calnexin by capturing vesicles from the ER exit site, which eventually attach to the ER for proliferation. Once BCV fusion with lysosomes is impaired or disrupted, it replicates in the ER and subsequently interacts with host autophagy-related proteins (Beclin1, ULK1, Atg14, IRE1α-UPR signaling axis), enhancing the autophagy of host cells phagocytosis and promoting the excretion of *Brucella* and the initiation of the replication cycle within newly infected cells [20,32,33]. On the basis of ensuring that host cells will not die due to infection for a long time, the ability of *Brucella* to replicate in the cell is particularly important and critical. When *Brucella* infects pregnant animals, it will directionally select to enter the trophoblasts colonies of placenta to form high-density bacterial colonies, which can easily lead to placentitis and even miscarriage in pregnant animals [34,35]. At present, the specific mechanism of *Brucella*’s entry into trophoblast cells is still not fully explained. In addition, *Brucella* replicates in the ER niche but not in all types of host cell, such as neutrophils [36,37].

### 2.2. The Dominant Carbon Source of Brucella—Erythritol

Generally speaking, in the presence of other sugars, erythritol is often the preferred carbon source for *Brucella* [38], but this is not enough to show that erythritol can completely determine the growth of *Brucella*. There seem to be inconsistent explanations for many phenomena here. First, not all mammalian placentas have sufficient erythritol. For example, human and mouse placentas are deficient in erythritol, but *Brucella* can still infect the placenta and cause inflammation [21,39,40]. Second, it has been reported that *B. ovis* and *B. canis* are known to be incapable of metabolizing erythritol but can cause reproductive tract infections and abortions in sheep and dogs, respectively [41,42]. Of course, there are also special cases where the S19 vaccine strain of *Brucella abortus* is inhibited by erythritol [43]. Therefore, these can only show that the presence of erythritol is an advantage rather than a necessary condition for the survival of the vast majority of *Brucella*. Although there is still no conclusive evidence for why *Brucella* prefers to colonize reproductive organs and tissues, the presence of large amounts of erythritol in the placenta of animals such as ruminants may be part of the molecular explanation for tropism [42,44]. However, recent work by Babier demonstrated that the presence of AR may be an important factor in the proliferation of *Brucella* [42]. AR is a key enzyme in polyol metabolism, which can catalyze the formation of various polyols under suitable conditions, including erythritol [45]. It suggests that erythritol in the placenta may be an alternative carbon source and not a necessary factor, and there may be an active synthetic pathway that can generate a variety of polyols, including erythritol [46]. Experiments have shown that large amounts of fructose and AR are found in the reproductive parts of pigs, cattle and rodents [47,48,49]. Therefore, the rich material reserves of the placenta and the existence of AR may provide a suitable living environment for *Brucella*, which requires further experimental verification. Erythritol metabolism of *Brucella* also affects its intracellular growth. Relying on the ery operon of *Brucella* (eryA, eryB, eryC, and eryD, etc.), erythritol is converted into erythrulose-1-phosphate by phosphorylation, followed by dehydrogenation and decarboxylation to finally generate diphosphate hydroxyacetone and carbon dioxide [50]. In general, erythritol metabolism is beneficial to the survival and proliferation of *Brucella* in trophoblast cells, which provides a prerequisite for *Brucella* to induce an inflammatory response in placental tissue.

### 2.3. CD98hc Damage

Some experiments have pointed out that the invasion of *Brucella* into trophoblast cells will cause damage to the function of trophoblasts, some of which are critical to the function of CD98hc [51]. CD98hc is a membrane surface glycoprotein that can mediate functions such as amino acid transport, integrin-dependent signal regulation, and cell fusion [52], but it is still unclear how CD98hc exerts amino acid transport and integrin functions in cells. In the human placenta model, CD98hc is required for the fusion of epithelial CTB to form multinucleated SYN or differentiation to form EVT. *Brucella* infection of CTB will indirectly affect the synthesis and function of SYN and EVT, thereby affecting the normal development of the placenta [51]. Additionally, the study of the mouse skin fibroblast model established by Anne et al. showed that the process of *Brucella* infection requires the uptake of CD98hc from host cells, which indicates that CD98hc may be involved in the adhesion and internalization process of *Brucella* and host cells [29]. However, it is still difficult to explain how the invasion of *Brucella* leads to the reduction of the expression level of CD98hc in host cells, and the mechanism by which *Brucella* affects the level of CD98hc still needs a lot of studies.

## 3. *Brucella* Mediates the Trophoblast Inflammatory Response

### 3.1. Effector Proteins: VceA and VceC

T4SS has long been identified as an important virulence factor of *Brucella* by the veterinary community, but there are still few studies on the T4SS effector protein and the interaction mechanism between *Brucella* T4SS and host cell structures [53]. VceA is one of the first substrates identified in *Brucella* T4SS, which is critical to the survival and replication of *Brucella* in host cells [54]. Autophagy is a common host protective strategy after *Brucella* infection of host cells, which can help host cells remove damaged proteins or organelles that are unfavorable to cells. It is currently known that *Brucella* VceA vaccine mutants can increase the level of autophagy in mouse and human trophoblast cell lines, thus affecting their own extracellular survival. Not only that, after infection of trophoblast cells by *Brucella* VceA mutant strain, it can induce the secretion of TNF-α and IL-1β, and finally inhibit the apoptosis of trophoblast cells [55]. Therefore, it is speculated that VceA may effectively reduce the autophagy level of host cells, the production of inflammatory factors and improve the probability of intracellular parasitic survival of *Brucella*.

VceC is another effector protein expressed by *Brucella* (Figure 2), which is conserved in all sequenced *Brucella* genomes [56]. It has been shown that *Brucella* abortion VceC can rely on the T4SS secretion system to transmigrate into mouse trophoblast cells and bind to the ER chaperone immunoglobulin GRP78 to disrupt ER structure and function, leading to ER stress and eventually cell death and placental inflammation [20,57]. The general inflammatory response is generated by the activation of PRRs, such as TLRs or NLRs, which detect pathogen infection or tissue damage. NOD1 and NOD2 play an important role in the inflammatory process triggered by VceC. They can respond to the UPR-induced IRE1α activation pathway. When the IRE1α pathway is activated, TRAF2 receives signals and translocates to the ER membrane, where it binds to RIP2 using NOD1 and NOD2 signaling molecules and activates the NF-κB pathway to recruit TAK1, triggering inflammatory responses [58,59,60]. This is followed by increased levels of IL-6, IL-12p40, IFNγ-γ and other inflammatory factors in host cells, which eventually lead to placenta inflammation and fetal death [61]. Similar results have been seen in previous studies, both in mice and ruminants, in vivo and in cell lines [19,62]. Thus, *Brucella* relies on T4SS to disrupt normal cellular pathways [63] and establish replication niches on the ER, which may be the trigger of trophoblast cell death caused by *Brucella*.

Moreover, VceC not only induces inflammation, but also regulates the apoptosis pathway of CHOP and affects the apoptosis of host cells, but a conclusion has not been unified. In a pregnant mouse model, deletion of VceC effectively reduced CHOP levels and increased the intracellular reproductive density of *Brucella*. VceC secretion causes ER stress, activates PERK pathway, induces the expression of Ddit3, and finally promotes the generation of CHOP in cells [20]. CHOP is a transcription factor of the CCAAT/ enhancer binding protein (C/EBP) family, which usually occurs during various physiological damage processes of cells, such as endoplasmic reticulum stress, DNA damage or pathogen invasion. Therefore, it is also one of the hallmarks of host cell initiation of apoptosis [64,65]. With the increase of CHOP levels, the host cells gradually dies and discharges *Brucella*, which promotes the next round of *Brucella* infection and proliferation. However, in the experiments in which *B. suis* attacked GTC [18], the emergence of VceC mutants promoted CHOP-induced apoptosis. The researchers treated the VceC deletion mutation and found that the mutant strain decreased the expression of GRP78 and enhanced the expression of CHOP, indicating that VceC enhanced the inflammatory response of host cells and inhibited apoptosis by interacting with GRP78. *Brucella* infection of trophoblast cells can cause ER and UPR, but inhibit host cell apoptosis, which may be necessary to ensure sustained proliferation of *Brucella* in host cells [66].

Although VceC activates IRE1 and other signaling pathways in trophoblast cells and causes discomfort to host cells, studies have found that the infection of VceC mutants and wild strains in host cells is similar, without significant difference, indicating that the loss of VceC does not reduce the intracellular replication ability of *Brucella* [61]. It was further shown that VceC is an important virulence factor regulating trophoblast apoptosis by activating ER stress and further manipulating UPR during *Brucella* infection, mainly inducing placental inflammation and affecting apoptosis, but not directly related to bacterial colonization [18,20,61]. Therefore, how VceC affects apoptosis and whether there are differences in the induction of CHOP among different species need to be further studied.

### 3.2. BtpA and BtpB

According to previous studies, BtpA and BtpB may be one of the factors that cause trophoblastic inflammation. There is some controversy here that, in the model of *Brucella abortus* infecting bovine trophoblast [67], the loss of BtpB impairs the ability of *Brucella abortus* to inhibit the pro-inflammatory reaction, but Fernández’s experiments [68] show that the loss of BtpA and BtpB has little effect on human trophoblast inflammation. BtpA and BtpB are *Brucella* Toll-like receptor domain-containing proteins that interfere with Toll-like receptor signaling to suppress host innate immune responses and modulate host cell inflammatory responses [67]. NAD is an important coenzyme involved in hundreds of enzymatic reactions, especially glycolysis, the TCA cycle, and mitochondrial oxidative phosphorylation. NAD has also been shown to be a key regulator of immune metabolism, acting as an important metabolic switch. In macrophages, increased levels of NAD have been shown to be associated with the activation and control of inflammatory responses, specifically involving the regulation of TNFα transcription in typically activated pro-inflammatory (M1) macrophages [69,70]. *Brucella* can utilize BtpA and BtpB effector proteins to reduce total NAD levels in host cells, which may help regulate cellular metabolism and signaling. Both translocation effectors, BtpA and BtpB, contain a TIR domain that has been shown to down-regulate innate immune signaling in specific in vitro differentiated mouse bone marrow-derived dendritic cells [71,72]. Indeed, targeting these *Brucella* effectors into the tonoplast or innate immune signaling platform may locally affect NAD levels that inhibit specific enzymatic responses. Interestingly, GAPDH, the first enzyme to use NAD in glycolysis, has been shown to be recruited to *Brucella*-containing tonoplasts and plays a role in intracellular replication [73]. Additionally, NAD levels are sensed by sirtuin proteins, which activate several signaling pathways, some of which are involved in immunomodulatory functions [74]. Not only that, BtpB also has the effect of suppressing NOD2 expression, which blocks NF-κB activation by stimulating MyD88 and inhibiting TLR signaling, which further inhibits the NF-κB inflammatory response to a certain extent [75]. Therefore, *Brucella* restricts host cell NAD levels and ATP energy metabolism through the expression of BtpA and BtpB effector proteins, which may help reduce the level of immune inflammation in host cells. Whether the effectors BtpA and BtpB of *Brucella* play consistent roles in trophoblast cells and in other cell types, this question needs more experimental investigation. [72].

### 3.3. Phosphoglucomutase

pgm is one of the important enzymes in *Brucella* LPS synthesis, which is responsible for the interconversion between glucose 6-phosphate and glucose 1-phosphate [76]. With the help of this gene, *Brucella* can synthesize the virulence factor, cyclic beta glucan [77], which is important for the intracellular infection cycle. Czibener’s experiments verified that the mutant strains that lost the pgm gene were unable to assemble and synthesize complete LPS, causing host cell immune recognition and being attacked [78]; the construction of the M5-90 mutant strain may also indicate that the changes caused by the deletion of pgm are also applicable to trophoblast cells. These studies point out that the presence of pgm is very important for the intracellular survival of *Brucella*. For pgm mutants, host cells can produce more inflammatory factors such as IFN-γ and IL-2, initiate protective immunity and reduce its intracellular survival rate [79]. Therefore, pgm can help *Brucella* evade the host immune system, which has far-reaching inspiration for the development of *Brucella* vaccines [80,81].

### 3.4. Trophoblasts Secrete High Mobility Group Box 1

Studies have shown that *Brucella* can infect trophoblast cells and induce them to release HMGB1 into the extracellular environment (Figure 3); The elevated level of HMGB1 can promote the immune system to recognize *Brucella* and produce an inflammatory response. Not only that, after using anti-HMGB1 antibodies to bind HMGB1 protein, it was found that both NADPH activity and the inflammatory response were inhibited. This indirectly indicates that HMGB1 is closely related to the occurrence of trophoblast inflammation [82]. In nature, HMGB1 is a ubiquitous and highly conserved protein. It is present in all types of cells and is one of the important family members of DAMPs [83]. Intracellular expression of HMGB1 is required for inflammatory responses during infertility and infection-related diseases. In all mammalian cells, the HMGB1 protein has a stable and complete physiological function, which is related to the activity of intracellular DNA substances and physiological activities, such as transcription, recombination, replication, and genome stability [84]. Outside of cells, HMGB1 is also involved in regulating various physiological activities of the body, such as cell death, immunity and inflammation. In addition to this, HMGB1 is also involved in the regulation of processes such as autophagy, apoptosis, and cancer [85]. Furthermore, the release of HMGB1 has the potential to activate TLRs and related receptors in bystander cells, further promoting the production of pro-inflammatory cytokines and the release of HMGB1 [82,84]. It has been experimentally demonstrated that in neutrophils, HMGB1 plays a vital role in regulating the production of ROS and the activation of NADPH oxidase [86]. HMGB1-induced ROS production can inhibit the growth of *Brucella* in host cells [87], which may be one of the reasons for the reduction in the number of viable bacteria in trophoblast cells. Therefore, after *Brucella* infection of trophoblast cells, the release of HMGB1 may stimulate the body to recognize pathogens, induce host cells to generate ROS and mediate inflammation, thereby inhibiting the growth of *Brucella* in trophoblast cells.

### 3.5. Secretion of Inflammatory Factors

In contrast to the mild granulomatous inflammation caused by infection of macrophages, *Brucella* can cause a strong inflammatory response and necrosis of trophoblast cells [88]. *Brucella* infection leads to an increase in the level of inflammation in the placental tissue of pregnant mice and the occurrence of abortion. The most intuitive data changes are the increase in TNF-α, IFN-γ, and normal expression and secretion of T cells after activation RANTES [89,90]. Byndloss et al. found that inhibition of endoplasmic reticulum stress or secretion of inflammatory factors could attenuate placental inflammation and improve fetal survival in mice [20]. This suggests that in placental tissue, ER stress-induced inflammation is inextricably linked to miscarriage, and inhibition of inflammation can prevent fetal loss [39,91]. T4SS plays a significant role in the entire process of *Brucella*-induced inflammation of the placenta. Among them, VceC, one of the T4SS effector proteins, induces the occurrence of ER stress in host cells, and TNF-α is greatly increased by ER stress stimulation and causes placental inflammation and fetal loss [20]. In addition to this, the placenta is a special organ with numerous cell types, including trophoblasts, macrophages, neutrophils, etc. In the case of *Brucella* infection, interactions between different cells may also promote the production of TNF-α, and the specific interaction mechanism is still unclear and needs to be further verified [61].

In previous experiments, it was found that canine placental trophoblast cells infected with *Brucella canis* were induced to secrete IL-8, IL-6 and RANTES during infection (Figure 3) [92]. A similar phenomenon occurs in trophoblast cells stimulated by factors produced by *Brucella canis*-infected monocytes. The secretion of IL-8 can make neutrophils move to the infection site [93]; RANTES is one of the important chemical components that can induce a variety of leukocytes to enter the infected tissue site [94], and the two may be involved in the placental tissue infiltration of *Brucella* infection. This suggests that canine trophoblast cells may produce pro-inflammatory cytokines in response to *Brucella* invasion on the one hand, and factors produced by infected monocytes and neutrophils on the other. Canine trophoblast cells respond to *Brucella* infection by secreting pro-inflammatory factors such as IL-8 and RANTES, and amplify this phenomenon, attracting inflammatory cells, such as neutrophils and monocytes, to the infection. When inflammatory cells come into contact with bacteria, they produce cytokines, such as inflammatory cytokines (Table 1), which further booster the trophoblast inflammatory response [92]. This is a similar phenomenon to previous studies examining the in vitro interaction between human trophoblast cell lines and phagocytes [51,68]. Furthermore, trophoblast cells stimulated with neutrophil-secreted factors had higher levels of inflammation than trophoblast cells infected with *Brucella* alone. It indicates that phagocytes play a key role in regulating the inflammatory response of placental tissue or abortion caused by *Brucella*.

The placenta provides a unique immune environment for the fetus during pregnancy, where immune tolerance is critical to avoid rejection of the fetus, which would otherwise be seen as a foreign body and attacked by the immune system. Bacterial infection of placental cells can change this by triggering a local inflammatory environment that can lead to pregnancy complications. In a recent report [68], increased secretion of IL-6, IL-8 and MCP-1 was observed in human cytotrophoblast Swan-71 cells infected with *B. abortus*. However, *B. melitensis* or *B. papionis* infection of JEG-3 or BeWo cells did not increase IL-6 or TNF-α levels. The discrepancy may be due to differences in differentiation states between trophoblast cell lines [95] or species dependent responses of *Brucella* to trophoblast cells. Further exploration is needed to verify its mechanism.

**Table 1 ijms-23-13417-t001:** Related cytokines involved in the infection of trophoblast cells by *Brucella*.

Inflammatory Factors Involved in the Placenta	Functions
TNF-α	Amplifies and coordinates pro-inflammatory signaling factors, mediates the expression of effector molecules, activates host resistance to pathogens after binding to TNFR-1, and enhances inflammatory responses in infected tissues [96].
IL-6	A multifunctional cytokine that is rapidly produced in response to infection and participates in host immune responses by stimulating acute-phase immune responses. It can play separate roles according to different immune conditions and is essential for the host’s innate immunity [97,98].
IL-8	IL-8, also known as CXCL8, is an important chemokine capable of recruiting a variety of phagocytic cells, co-mediating inflammatory responses by responding to other pro-inflammatory cytokines and is an important determinant of the differentiation of inflammatory sites [93].
IL-12p40	IL-12, together with IL-12p35, is an important factor in the protective immunity caused by *Brucella* infection. In the inflammatory response, IL-12p40 secreted by inflammatory cells may determine the genotype differences between *Brucella* hosts to a certain extent [99].
RANTES	A variety of leukocytes can be recruited in the inflammatory site, causing inflammatory infiltration and participating in the pathological process of *Brucella*-induced abortion. It is one of the important pro-inflammatory signaling molecules between trophoblasts and phagocytes. Often associated with IFN-γ pro-inflammatory responses in TH1 immune responses [92].
IFN-γ	It is an important part of the TH1 immune response and is secreted by activated TH1-secreting cells. It enhances the killing effect on *Brucella* by activating macrophages and promotes the production of inflammation in infected tissues [100].
IL-1β	In *Brucella* infection, IL-1β is required for the secretion of intracellular ROS in host cells. Their induction is required for the production of various chemokines and adhesion molecules, and neutrophils and monocytes are stimulated by IL-1β to enhance phagocytic activity. The maturation of IL-1β is regulated by the inflammasome, and the two are closely related [101,102].
MCP-1	As one of the pro-inflammatory cytokines, macrophages can be recruited at the infection site, so that macrophages tend to move to the infection site, thereby increasing the local inflammatory level, which is of great significance in maintaining inflammation and improving host immunity [103].
IL-2	It is a cytokine produced by activated T cells and plays an important role in immunity. It can be either an autocrine growth factor for T cells or a paracrine growth factor for NK cells. The secretion of IL-2 also plays an important role in the generation and maintenance of immune tolerance [104].
IL-10	Different from pro-inflammatory cytokines such as TNF-α and IFN-γ, IL-10 is usually produced by Th2 cells, monocytes, macrophages and its leading role is to inhibit inflammation and Th1-dependent immune responses. Plays a hindering role in abortion caused by bacteria and facilitates *Brucella* escape from phagolysosome fusion [105].

## 4. Survival Mechanism of *Brucella* in Trophoblast Cells

The proliferation of *Brucella* after entering the trophoblast cells is not rapid. The placenta is one of the most suitable organs for the proliferation of *Brucella* [106,107], and its abundance of erythritol can be used as the preferred carbon source for *Brucella*, and it will directionally drive *Brucella* to transfer to the placenta. After obtaining sufficient energy reserves and a stable environment, *Brucella* proliferates in enormous quantities. During the last trimester of gestation in venomous ruminants, *Brucella* begins to replicate extensively, with clear colonies visible in the placenta. *Brucella* rapidly proliferates and replicates in these host cells, leading to disruption of placental integrity and infection of the developing fetus, leading to miscarriage or weak litters. Anderson et al. dissected placental tissue from the cadavers of infected goats [62], and isolated *B. abortus*, followed by microscopic observation and analysis. The results show that *Brucella* replicates in the cellular compartment associated with the rough ER in the placental trophoblast. The high-efficiency replication brought about by this special environment causes placental tissue inflammation, inflammatory cell infiltration, trophoblast cell necrosis, and chorionic alanine membrane ulceration, which eventually leads to miscarriage [108,109]. This is an efficient way for the progeny of *Brucella* to spread to new hosts [110]. It is worth mentioning that progesterone produced by the placenta of pregnant mice can effectively inhibit the NF-κB inflammatory pathway caused by *Brucella* infection of trophoblast cells and reduce the level of placental inflammation [111], which is the reason for the decreased ketone level. In addition, progesterone can promote the expression of NOX2 in infected trophoblast cells to increase ROS production [112]. Experiments have shown that the secretion of progesterone from mouse placenta could inhibit the growth of *B. abortus* in trophoblast cells. Progesterone treatment may be a means to prevent placentinitis and miscarriage, but progesterone receptors are present in a large number of cells. Poor specificity in tissue is one of the limiting factors hindering the development of such therapies [111].

*Brucella* has now evolved a successful strategy to avoid recognition by the host immune system and promote its survival and replication, thereby establishing a persistent infection in the host [113]. In the early stage of invasion, *Brucella* uses the low negative charge of LPS to prevent the recognition and binding of a variety of disease-resistant molecules with high activity, which makes it impossible for complement to contact and react with *Brucella* [114]. When the structure of LPS was modified or destroyed, we found that *Brucella* mutants lacked O chains, damaged core oligosaccharides, or failed lipid A could not avoid complement binding and thus be eliminated [115]. The core site of LPS is the key factor that constitutes the virulence of *Brucella*. Not only that, the PAMP components (lipoproteins, ornithine-related lipids, and flagellin, etc.) of *Brucella* are very weak innate immune activators, which can well evade the killing of the host cell immune system [35]. All these provide conditions for *Brucella* to safely reach the host cells. After invading the body, it exists in phagocytes and trophoblast cells, and establishes a replication niche suitable for survival and replication in the ER. The establishment of a replicative niche is related to the long-term replicative survival cycle of *Brucella* [110]. *Brucella* can survive in this way even when the number of viable bacteria is extremely low. The reason why *Brucella* can stably and safely colonize trophoblast cells is that it effectively hides its own pathogen-associated molecular patterns, causing the host immune surveillance system to fail, and then secretly spread throughout the reticuloendothelial system [116].

The effect of *Brucella* infection on trophoblast cells is a complex physiological process. It is still unclear how *Brucella* carries out material consumption and energy utilization in trophoblast cells, but it is known that in the process of chronic *Brucella* infection of a large number of macrophages, host cells gradually change from glucose oxidative metabolism to fatty acid oxidative metabolism. It is beneficial to improve the utilization of glucose by *Brucella*, thereby promoting the intracellular replication of *Brucella* [117]. Autophagy and initiation of tissue inflammation are notable features of *Brucella* infection of trophoblast cells. When *Brucella* attaches to the rough ER intracellularly, it binds to a variety of autophagy-related factors and then upregulates the autophagy level of host cells and inhibits the apoptosis program. Persistent reproduction provides favorable conditions. When trophoblast cells are stimulated by foreign pathogens, they secrete inflammatory factors (IL-6, IL-8, MCP-1, etc.) and HMGB1 and other substances to promote the placental inflammatory response and induce neutrophils and monocytes to move to the placental tissue to achieve the purpose of removing pathogenic factors. In this process, *Brucella* inhibits the inflammatory response of host cells through different pathways. The presence of VceA can effectively affect the secretion of inflammatory factors, such as TNF-α and IL-1β, in host cells and reduce the level of autophagy; BtpA and BtpB can interfere with host cell signal regulation and transduction, prevent the spread of cytokines, and lead to intracellular NAD levels and ATP metabolism, which further slows down the rate of inflammatory responses. These may be necessary conditions to ensure that *Brucella* can sustainably reproduce and replicate in trophoblast cells.

Compared with general cell infection, the infection mechanism of *Brucella* in trophoblast cells is more complicated, which not only affects the metabolism and function of host cells, but also involves the placenta, a special tissue, and related physiology such as fetal development. At present, there are still few studies on the specific mechanism of *Brucella*-induced trophoblast inflammatory response and immune response, and many issues need to be further studied and discussed.

## 5. Conclusions

In general, infection of trophoblast cells by *Brucella* can lead to increased secretion of inflammatory cytokines by placental tissue, resulting in phagocyte colonization and aggregation phenomenon, leading to clinical symptoms of inflammation and abortion. The placenta, a reproductive organ connecting the mother and fetus, is an intermediate in the communication of information substances between the two. The occurrence of inflammatory responses is the result of the joint participation of many types of cells. This is more complicated than other types of infection events. Although the pro-inflammatory mechanisms are similar in some respects, this may not hold true for all infection scenarios. At present, the infection of trophoblast cells by *Brucella* is still insufficiently studied and certain physiological changes may vary depending on the model in which a single species infects a single host. Therefore, to analyze the systemic *Brucella* pro-inflammatory mechanism as a whole still needs a lot of research.

## Abbreviation

AbbreviationDefinitionATF6activation of transcription factor 6Atg14autophagy-related 14ARaldose reductaseATPAdenosine triphosphate
*B. melitensis*

*Brucella melitensis*

*B. abortus*

*Brucella abortus*

*B. suis*

*Brucella suis*

*B. ovis*

*Brucella ovis*

*B. canis*

*Brucella canis*

*B. neotomae*

*Brucella neotomae*

*B. ceti*

*Brucella ceti*

*B. papionis*

*Brucella papionis*

*B. inopinata*

*Brucella inopinata*

*B. pinnipedialis*

*Brucella pinnipedialis*

*B. microti*

*Brucella microti*

*B. vulpis*

*Brucella vulpis*
BCV*Brucella*-containing vacuolesCD98hcCD98 heavy chainCHOPC/EBP homologous proteinCXCL8C-X-C motif ligand 8CTBcytotrophoblastsDAMPsdamage-associated molecular patternsERendoplasmic reticulumEEA1Early Endosomal Antigen 1EVTextravillous trophoblastsGRP78Glucose-Regulated Protein 78GTCgoat trophoblast cellsGAPDHglyceraldehyde 3-phosphate dehydrogenaseHMGB1High Mobility Group Box 1IRE1inositol-requiring enzyme 1IFN-γinterferon gammaILinterleukinLAMP-1Lysosomal Associated Membrane Protein 1LPSlipopolysaccharideMyD88myeloid differentiation primary response gene 88MCP-1monocyte chemotactic protein 1NODnucleotide-binding oligomerization domainNF-κBnuclear factor kappa BNKnatural killerNOX2nicotinamide adenine dinucleotide phosphate oxidase 2NADPHReduced nicotinamide adenine dinucleotide phosphateNLRsNOD-like receptorsPAMPpathogen associated molecular patternPERKprotein kinase RNA-like ER kinasePRRspattern recognition receptorspgmphosphoglucomutaseROSreactive oxygen speciesRANTESregulated upon activation normal T-cell expressed and secretedRIP2receptor-interacting serine/threonine-protein kinase 2SYNsyncytiotrophoblastsT4SSType IV secretion systemTNF-αtumor necrosis factor-αTLRsToll-like receptorsTRAF2TNF receptor associated factor 2TAK1TGF-β-activated kinase 1TCAtricarboxylic acidTIRToll/interleukin-1 receptorUPRunfolded protein responseULK1Unc-51-like kinase 1

## Figures and Tables

**Figure 1 ijms-23-13417-f001:**
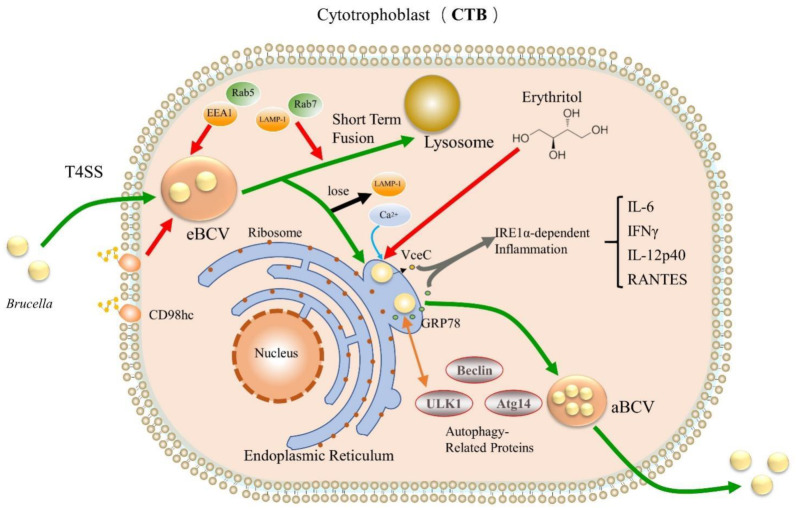
Intracellular process of *Brucella* invading CTB. Green arrows indicate the infection pathway of *Brucella*; red arrows indicate adsorption, acquisition or capture; gray arrows indicate co-cause and mediation; orange arrows indicate interactions; blue arrows indicate binding; black arrows indicate loss.

**Figure 2 ijms-23-13417-f002:**
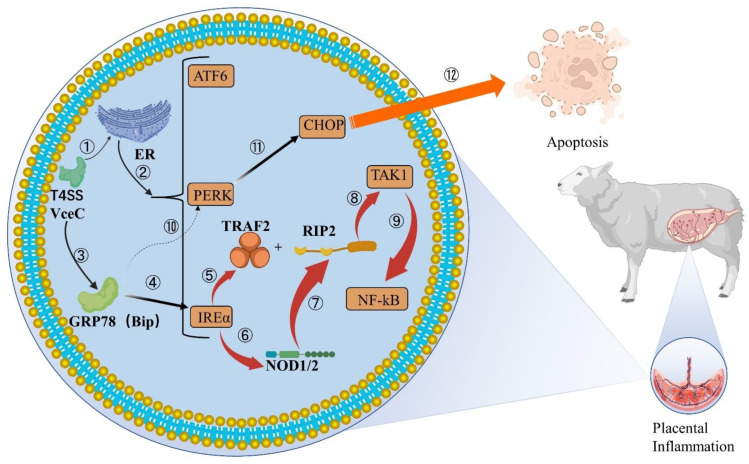
VceC mediates IREα inflammatory responses and apoptosis in cells. ①②: VceC ectopic to rough ER through *Brucella* T4SS. RER is stimulated by VceC and causes ER stress to increase unfolded products, and prolonged stress leads to UPR, activation of ATF6, PERK and IREα metabolic pathways. ③④: VceC binds to the GRP78 chaperone at the ER to induce the IREα inflammatory pathway. ⑤–⑨: IREα is activated and recruits TRAF2 to the ER membrane and triggers NOD1/2 homo-oligomerization and activation. TRAF2 binds and k63-polyubiquitinates RIP2, both of which are involved in NOD1/2 signaling to mediate the collection of TAK1, ultimately activating NF-κB, thus increasing the levels of inflammatory factors such as IL-6, IL-12p40, and causing the placental occurrence of inflammation. ⑩–⑫: VceC binding to GRP78 affecting the changes of the PERK pathway in host cells is still controversial. ER stress induces another important pathway, the PERK apoptosis pathway, which induces the expression of CHOP by gene Ddit3, which eventually leads to apoptosis.

**Figure 3 ijms-23-13417-f003:**
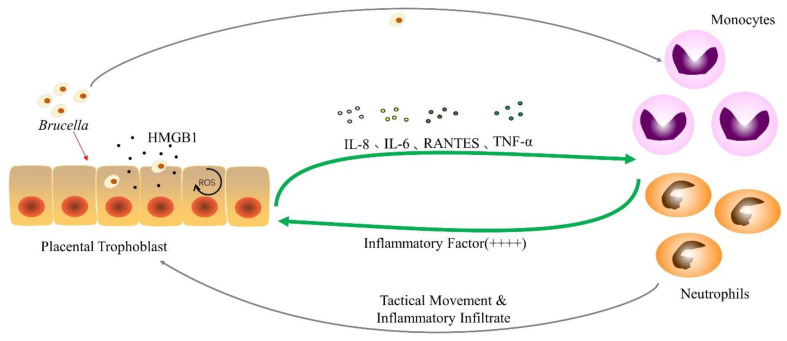
The interaction between *Brucella* and phagocytic cells after infection of trophoblast cells. After the trophoblast cells are attacked by *Brucella*, they begin to produce HMGB1 and excrete HMGB1, increasing the concentration of ROS in response to changes in the intracellular environment and secrete cytokines such as IL-6, IL-8, RANTES and TNF-α. Monocytes and neutrophils move to the placental infection site and secrete a large number of inflammatory factors after receiving the stimulation of *Brucella* and cytokines, causing placental inflammatory infiltration.“++++” indicates an increase in the degree of inflammatory response, which is a positive feedback of inflammatory cells to trophoblast cell signals. The specific inflammatory secreted factors are shown in Table 1.

## Data Availability

Not applicable.

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
