# Peer review of "Inflammatory Mechanism of Brucella Infection in Placental Trophoblast Cells"

_ijms, 2022, doi:10.3390/ijms232113417_

Round 1
Reviewer 1 Report
This review introduces the research progress of Brucella infection in placental trophoblasts in detail from the aspects of Brucella invasion, transport, carbon source utilization, CD98hc, and mediated inflammatory response in trophoblast cells. The content of the review is novel and comprehensive, which is valuable for understanding the pathogenic mechanism of Brucella infection. However, this paper suffers a lot of flaws in format, grammar, and logic, so the authors should modify the following points in detail.
1.The format of the references in the text, such as [1] (1), has never been seen before. Therefore, the authors should modify the reference format as required by the IJMS.
2. The language needs to be improved. There are many grammatical errors, wrong expressions, wrong formats, and colloquial expressions in the whole manuscript.
3.Multiple misuses of punctuation in this review, e.g., a period in an incomplete sentence. (page 12, line 5, page 12, line 17…)
4. In the introduction section, the author mentioned that B. abortus, B. suis, and B. canis also have the same pathogenic potential in humans, which is a wrong depiction. As we all know, the virulence of the three trains is different, and the pathogenic is different, too.
5.The second title does not match the content of the second section. The title is about Brucella invasion, and the second part is about Brucella intracellular processes, so it is suggested to change the title or the contents.
6. The title of section 3 is ‘Brucella Mediates Trophoblasts Inflammatory Response.’ VceA and VceC have been identified to participate in the inflammatory response in trophoblasts during Brucella infection. However, there was no evidence to prove that BtpA or BtpB could manipulate inflammatory responses in trophoblasts. So, this point should be distinguished.
Author Response
1. The format of the references in the text, such as [1] (1), has never been seen before. Therefore, the authors should modify the reference format as required by the IJMS.
Response: I really appreciate your constructive proposal, but I'm sorry that I didn't completely understand the comments. Before submitting the article, I referred to the format requirements of the IJMS journal and edited it on the provided Word template. References were also written on the EndNote platform. So I checked it and found no duplicate marks. Reviewer 3 also mentioned this issue, so, it may require further communication.
2. The language needs to be improved. There are many grammatical errors, wrong expressions, wrong formats, and colloquial expressions in the whole manuscript.
Response: Thank you very much for your correction. I reread the full text, sorted out relevant knowledge and corrected the mistakes I could find. Thank you again.
3.Multiple misuses of punctuation in this review, e.g., a period in an incomplete sentence. (page 12, line 5, page 12, line 17…)
Response: Thank you for your comments. Your comments are very helpful to the revision of the article. I rearranged all punctuation marks to make the sentences more fluent.
4. In the introduction section, the author mentioned that B. abortus, B. suis, and B. canis also have the same pathogenic potential in humans, which is a wrong depiction. As we all know, the virulence of the three trains is different, and the pathogenic is different, too.
Response: Thank you for your correction. There are some cognitive errors here, and I changed the original sentence into “For humans, B. melitensis has the strongest pathogenicity, followed by B. abortus, B. suis and B. canis”.
5.The second title does not match the content of the second section. The title is about Brucella invasion, and the second part is about Brucella intracellular processes, so it is suggested to change the title or the contents.
Response: Thank you for your correction. In the second part, we would like to explain the physiological process after brucella invades trophoblast cells, so we will change the title to “Biological Process of Brucella in Trophoblast Cells”.
6. The title of section 3 is ‘Brucella Mediates Trophoblasts Inflammatory Response.’ VceA and VceC have been identified to participate in the inflammatory response in trophoblasts during Brucella infection. However, there was no evidence to prove that BtpA or BtpB could manipulate inflammatory responses in trophoblasts. So, this point should be distinguished.
Response: Thank you very much for this crucial proposal. Although there is currently no definitive evidence that BtpA and BtpB are involved in the inflammatory response in trophoblast cells, we believe that BtpA and BtpB may be an inflammatory regulator for the invasion of trophoblast cells by Brucella based on the available reference “Early transcriptional responses of bovine chorioallantoic membrane explants to wild type, DeltavirB2 or DeltabtpB Brucella abortus infection” and “Proinflammatory Response of Human Trophoblastic Cells to Brucella abortus Infection and upon Interactions with Infected Phagocytes”. We wanted to dissect out the physiological roles of BtpA and BtpB intracellularly and to derive from them parallels that would be true for trophoblast cells. So we added some more in paragraphs.

Reviewer 2 Report
Nice article
Author Response
We appreciate your valuable suggestions.
Reviewer 3 Report
The paper tries to be an overview of trophoblast cells infection by Brucella and its interaction with the placenta in the foetal development in brucellosis infections. The revision of pathogenesis of Brucella infections is well done and very well documented, but not very didactic and some sections difficult to follow. Specially is very difficult the section 3.1. Effector proteins: VceC (page 5 ).
Reading the article, I don't quite understand what the authors' objective is for its review. There are already several articles of bibliographic reviews published about Brucella pathogenesis, some of these articles are cited by the authors, and others have been omitted, in spite the same authors published them (Jiao, H.; Zhou, Z.; Li, B.; Xiao, Y.; Li, M.; Zeng, H.; Guo, X.; Gu,G. The Mechanism of Facultative Intracellular Parasitism of Brucella. Int. J. Mol. Sci. 2021, 22, 3673.https://doi.org/10.3390/ijms22073673; Xiong, X.; Li, B.; Zhou, Z.; Gu, G.; Li, M.; Liu, J.; Jiao, H. The VirB System Plays a Crucial Role in Brucella Intracellular Infection. Int. J. Mol. Sci. 2021, 22, 13637.https://doi.org/10.3390/ijms222413637). In my opinion the paper should be justifier its publication and discuss the review the authors did.
Moreover, also in my opinion, the authors should pay special attention when a result is obtained in an article in vivo or in vitro infections, and different in B. abortus (smooth Brucella) than in B. canis (rough Brucella) and different if they are done in mice or in calves. I think that those things worth mentioning every they are cited, because one fact does not have the same meaning in differ conditions.
Conclusions are conducted in several sections of the article but not well summarised in point 5. Conclusion. In my opinion this section should be rewritten and done it more elaborated. In general, contributions by the authors who make a discussion of the results are missing in the article.
The article is full of abbreviations and although many are indicated, their meaning is omitted in many others (T4SS, GRP78, RIP2, PERK, CHOP, NAD, TNF etc). Perhaps making a list of abbreviations would facilitate the reading of the article.
The references are cited doubled with [ ] and ( ) from page 1 to page 11 line 18. The references are the same number into [ ] and ( ) but in Table 1 and lines 1 to 18 in page 11 they have different number.
All Brucella species should be written in cursive letter in the text article, Table 1 and in the references too, in spite they were written abbreviated (B. abortus etc). All figures and tables should be name in the text of article and in this paper Figure 2 and Table 1 are not named.
Some cited references are electronic paper and then no pages in they when are cited but some electronic number should named if doi number is not name (ie. Reference 17: Int. J. Mol. Sci. 2019, 20, 4104; doi:10.3390/ijms20174104; ie. Reference 18: MBio 2019, 10:e01538-19).
Is “Brucella papionis” not “B. papinions” and “rough ER” not “rough RER”
Author Response
The paper tries to be an overview of trophoblast cells infection by Brucella and its interaction with the placenta in the foetal development in brucellosis infections. The revision of pathogenesis of Brucella infections is well done and very well documented, but not very didactic and some sections difficult to follow. Specially is very difficult the section 3.1. Effector proteins: VceC (page 5 ).
Response: Thank you very much for your comments. I have rearranged the relevant content about VceC to make the paragraph more understandable.
Reading the article, I don't quite understand what the authors' objective is for its review. There are already several articles of bibliographic reviews published about Brucella pathogenesis, some of these articles are cited by the authors, and others have been omitted, in spite the same authors published them (Jiao, H.; Zhou, Z.; Li, B.; Xiao, Y.; Li, M.; Zeng, H.; Guo, X.; Gu,G. The Mechanism of Facultative Intracellular Parasitism of Brucella. Int. J. Mol. Sci. 2021, 22, 3673.https://doi.org/10.3390/ijms22073673; Xiong, X.; Li, B.; Zhou, Z.; Gu, G.; Li, M.; Liu, J.; Jiao, H. The VirB System Plays a Crucial Role in Brucella Intracellular Infection. Int. J. Mol. Sci. 2021, 22, 13637.https://doi.org/10.3390/ijms222413637). In my opinion the paper should be justifier its publication and discuss the review the authors did.
Response: Thanks for the questions you addressed, we prepared this article with the aim of providing a better review of the pathogenic mechanisms of Brucella in trophoblast cells. Brucellosis remains a big problem as a zoonotic infectious disease and although there are currently many other studies on Brucella, it is still lacking in terms of pathogenicity to trophoblast cells and many mechanistic questions remain unanswered. The greatest impact of brucellosis is on chronic infection of the host, whose invasion into the reproductive system is difficult to treat. Further understanding of the infection mechanism of Brucella on the reproductive system is helpful for future research and treatment of the disease. Previously we summarized how Brucella parasitize inside the cell and T4SS virB, an important virulence factor of Brucella, which is a discussion of different perspectives of brucellosis. Combining these parts, I think it will be more comprehensive to know Brucella which has a bedding effect for future studies.
Moreover, also in my opinion, the authors should pay special attention when a result is obtained in an article in vivo or in vitro infections, and different in B. abortus (smooth Brucella) than in B. canis (rough Brucella) and different if they are done in mice or in calves. I think that those things worth mentioning every they are cited, because one fact does not have the same meaning in differ conditions.
Response: Thank you very much for your points, which I quite agree with. So I reorganized the article and mentioned it in some related content. for example, “It has been shown that Brucella abortion VceC can rely on the T4SS secretion system to transmigrate into mouse trophoblast cells and bind to ER chaperone immunoglobulin GRP78 to disrupt ER structure and function, leading to ER stress and eventually cell death and placental inflammation.”, “According to previous studies, BtpA and BtpB may be one of the factors that cause trophoblastic inflammation. There is some controversy here that, in the model of Brucella abortus infecting bovine trophoblast, the loss of BtpB impairs the ability of Brucella abortus to inhibit the proinflammatory reaction, but Fernández's experiment shows that the loss of BtpA and BtpB has little effect on human trophoblast inflammation .” or “Byndloss et al found that inhibition of endoplasmic reticulum stress or secretion of inflammatory factors could attenuate placental inflammation and improve fetal survival in mice.”.
Conclusions are conducted in several sections of the article but not well summarised in point 5. Conclusion. In my opinion this section should be rewritten and done it more elaborated. In general, contributions by the authors who make a discussion of the results are missing in the article.
Response: Thanks for your proposal, I have changed the final conclusion to “In general, infection of trophoblast cells by Brucella can lead to increased secretion of inflammatory cytokines by placental tissue, resulting in phagocyte colonization and aggregation phenomenon, leading to clinical symptoms of inflammation and abortion. The placenta, a reproductive organ connecting the mother and fetus, is an intermediate in the communication of information substances between the two. The occurrence of inflammatory responses is the result of the joint participation of many types of cells. This is more complicated than other types of infection events. Although the proinflammatory mechanisms are similar in some respects, this may not hold true for all infection scenarios. At present, the infection of trophoblast cells by Brucella is still insufficiently studied and certain physiological changes may vary depending on the model in which a single species infects a single host. Therefore to analyze the systemic Brucella proinflammatory mechanism as a whole still needs a lot of research.”
The article is full of abbreviations and although many are indicated, their meaning is omitted in many others (T4SS, GRP78, RIP2, PERK, CHOP, NAD, TNF etc). Perhaps making a list of abbreviations would facilitate the reading of the article.
Response: Thanks for your suggestions, I have edited most of the abbreviations in the text into a list and placed them at the end of the text.
The references are cited doubled with [ ] and ( ) from page 1 to page 11 line 18. The references are the same number into [ ] and ( ) but in Table 1 and lines 1 to 18 in page 11 they have different number.
Response: I really appreciate your constructive proposal, but I'm sorry that I didn't completely understand the comments. Before submitting the article, I referred to the format requirements of the IJMS journal and edited it on the provided Word template. References were also written on the EndNote platform. So I checked it and found no duplicate marks. Reviewer 1 also mentioned this issue, so, it may require further communication.
All Brucella species should be written in cursive letter in the text article, Table 1 and in the references too, in spite they were written abbreviated (B. abortus etc). All figures and tables should be name in the text of article and in this paper Figure 2 and Table 1 are not named.
Response: I gratefully acknowledge your suggestions, and I italicize Brucella species in the text and indicate Figure2 and table1.
Some cited references are electronic paper and then no pages in they when are cited but some electronic number should named if doi number is not name (ie. Reference 17: Int. J. Mol. Sci. 2019, 20, 4104; doi:10.3390/ijms20174104; ie. Reference 18: MBio 2019, 10:e01538-19).
Response: Thanks for your correction, I have rechecked the references and corrected some of the formatting errors.
Is “Brucella papionis” not “B. papinions” and “rough ER” not “rough RER”
Response: Thanks for your comments, I have revised the contents in error.

Round 2
Reviewer 3 Report
"Brucella" should be in italics always, also in lines: 155, 175, 178, 186, 187, 206, 213, 219, 225, 227, 247, 248, 275, 391, 448, 456, 458 and in References as all Brucella species.
"Brucella canis" should be in italics too at line 336
Author Response
"Brucella" should be in italics always, also in lines: 155, 175, 178, 186, 187, 206, 213, 219, 225, 227, 247, 248, 275, 391, 448, 456, 458 and in References as all Brucella species. "Brucella canis" should be in italics too at line 336.
Response: We have carefully revised it according to your comments, Thank you very much.